# A Flexible Hybrid BCH Decoder for Modern NAND Flash Memories Using General Purpose Graphical Processing Units (GPGPUs)

**DOI:** 10.3390/mi10060365

**Published:** 2019-05-31

**Authors:** Arul Subbiah, Tokunbo Ogunfunmi

**Affiliations:** Department of Electrical Engineering, Santa Clara University, 500 El Camino Real, Santa Clara, CA 95053, USA; togunfunmi@scu.edu

**Keywords:** BCH, decoder, iBM, GPU, hybrid, flash memory, Galois field, CUDA

## Abstract

Bose–Chaudhuri–Hocquenghem (BCH) codes are broadly used to correct errors in flash memory systems and digital communications. These codes are cyclic block codes and have their arithmetic fixed over the splitting field of their generator polynomial. There are many solutions proposed using CPUs, hardware, and Graphical Processing Units (GPUs) for the BCH decoders. The performance of these BCH decoders is of ultimate importance for systems involving flash memory. However, it is essential to have a flexible solution to correct multiple bit errors over the different finite fields (GF(2${}^{m}$)). In this paper, we propose a pragmatic approach to decode BCH codes over the different finite fields using hardware circuits and GPUs in tandem. We propose to employ hardware design for a modified syndrome generator and GPUs for a key-equation solver and an error corrector. Using the above partition, we have shown the ability to support multiple bit errors across different BCH block codes without compromising on the performance. Furthermore, the proposed method to generate modified syndrome has zero latency for scenarios where there are no errors. When there is an error detected, the GPUs are deployed to correct the errors using the iBM and Chien search algorithm. The results have shown that using the modified syndrome approach, we can support different multiple finite fields with high throughput.

## 1. Introduction

NAND flash memories are widely used in many electronic devices. These devices face reliability issues because of the densely-populated memory cells [1]. In fact, the 3D method used to manufacture flash memories, discussed in detail by Spinelli et al. [2], enforces the necessity to have high throughput error correction techniques. Bose–Chaudhuri–Hocquenghem (BCH) codes [3] are the most common error correction mechanisms for flash memory devices and other digital communications like optical networks. The increasing efficiency and throughput of the flash memory systems have drawn researchers to provide highly-efficient BCH decoders. The three major categories of the BCH decoders proposed are Central Processing Units (CPUs), hardware circuits, and Graphical Processing Units (GPUs). Cho proposed an efficient CPU-based implementation in [4], and Poolakkaprambil discussed multi-bit error using Hamming, BCH, and Low-Density Parity Check (LDPC) codes in [5]. Later, Lee et al. proposed a high throughput hardware architecture in [6]. Moreover, Zhang discussed different hardware implementation techniques in [7]. Qi et al. [8] proposed a GPU-based BCH decoder; later, we proposed an efficient algorithm for BCH decoders using GPUs in [9]. In addition to the requirement of high throughput, modern BCH decoders are required to support multiple bit error correction across various block sizes, which is the focus of this paper. Technology scaling has rendered the ability to integrate multiple GPUs within a System On Chip (SOC), which has enabled researchers to use GPU for non-graphical applications. In fact, the term General Purpose Graphical Processing Unit (GPGPU) refers to the application of GPU for nongraphical applications. We use the term GPU instead of GPGPU since these terms are interchangeable in practice. Streaming Multiprocessors (SMs) are the building blocks of these GPUs, which has multiple CPUs within them. Each of the instantiated SMs is capable of handling multiple threads, which are scheduled by a warp scheduler. Therefore, we need an exclusive compiler like the Computer Unified Device Architecture (CUDA) C [10] software to program these SMs. The CUDA software creates the necessary grid of kernel routines, which in turn create the same instruction that operates on a different data path; this technique is referred to as the single instruction multiple data (SIMD) stream. The kernel subroutines are executed across multiple cores and in a multiple thread fashion. The GPU-based BCH decoders [9] are flexible, and they can support multiple BCH block sizes.

We have organized this paper as follows: Section 2 discusses the background and previous works. Section 3 describes our proposed hybrid method using GPUs and hardware design. Section 4 presents the results observed, and we conclude in Section 5.

## 2. Background

BCH codes are cyclic block codes encoded by the generator polynomial g(x) over the GF(2). The roots of this polynomial equation reside in the extended field, also known as the splitting field, $GF(2^{m})$. Let $\phi_{i}\left(x\right)$ be the minimal polynomial of an arbitrary element $\beta^{i}$, then the generator polynomial for BCH code with *t* error correction capability is given by the following equation:(1)$$g\left(x\right)=LCM(\phi_{1}\left(x\right),\phi_{2}\left(x\right),\ldots ,\phi_{2t}\left(x\right))$$

Narrow sense BCH codes use primitive element $\alpha^{i}$ for the minimal polynomial with *i* starting from one. For simplicity, the narrow sense BCH code decoder is discussed and implemented in this paper, and it could be easily extended for other general BCH codes [3]. The parity bits are then generated using the equation p(x)=m(x)modg(x), and these parity bits are concatenated to form the message polynomial m(x). This concatenation is given as:(2)$$c\left(x\right)=m\left(x\right)\;\cdot \;x^{deg(g(x))}+m\left(x\right)\;mod\;g\left(x\right)$$

The generated parity bits are then stored in the spare area allocated in the page within the flash memory device. In general, the hard decision BCH decoder has three steps in the decoding process: syndrome generation, key-equation solver, and an error locator.

### 2.1. Encoder

The main issue when using large BCH codes, i.e., *t* greater than 30, is the fan-out issue created by implementing the Linear Feedback Shift Register (LFSR) method of the generator polynomial. Parhi has addressed this fan-out issue by breaking down the LFSR register into multiple cascaded LFSRs by realizing the circuit in the Z-domain [11]. Hao addressed the same issue by using the Chinese Remainder Theorem (CRT) method [12], but this method requires more computation on the encoder and is applicable for encoders that have *t* higher than 100. Later, Tang et al. proposed a hybrid approach for long BCH encoders that is area efficient [13]. The authors of this paper had proposed an area efficient method by sharing the hardware between the encoder and syndrome generator [14].

### 2.2. Decoder

The BCH decoders can be categorized as hard decision and soft decision decoders [15,16]. These decoders’realization can be broadly classified into three categories: Central Processing Unit (CPU) [4], Very Large Scale implementation (VLSI) [17], and GPU implementation [8]. Various hardware implementations of BCH decoders were discussed by Zhang [7]. BCH decoders can be categorized by the place of the decoders. The decoders can be either located on-chip within memory device [18] or outside the memory device [19]. The focus of this paper is on the decoder being outside the memory device.

The syndrome generator is the first step of the BCH decoding process [6,20]. The syndromes $S_{i}$ of the received vector r(x) are given as:(3)$$S_{i}=r\left(\alpha^{i}\right)$$

In other words, the syndrome generator checks if the received code vector $r\left(x\right)=r_{n-1}x^{n-1}+\ldots +r_{1}+r_{0}$ has the roots as $\alpha^{1}$, $\alpha^{2}$, …, $\alpha^{2t}$. If so, then there are no errors in the received code vector. In the case of an error, the key-equation solver and the error locator steps are executed. For *t* bit error correction on a narrow sense BCH code, it is sufficient to find *t* syndromes, because the elements of a conjugacy class have the same minimal polynomial $\phi_{i}\left(x\right)$. We have discussed an alternate approach to share the syndrome generator and encoders in [14]; Figure 1 depicts the area sharing between the encoder and the syndrome generator presented in [14]. This method requires separate error protection to the parity bits, and one proposal is to use a Single-Level Cell (SLC) for the parity bits to reduce error probability.

An error locator polynomial Λ(x), which has dependency on the error location, gives a hint about the error location, and it is given by the equation:(4)$$\Lambda \left(x\right)=\sum_{i=0}^{t}\Lambda_{i}x^{i}=\left(1-X_{1}x\right)\left(1-X_{2}x\right)\ldots \left(1-X_{t}x\right)$$
where $X_{i}$ represents the error location of the vector r(x). The key equation:(5)S(x).Λ(x)=Ω(x)mod2t
shows the relationship between the error locator polynomial and the error evaluator polynomial; moreover, Newton identities [3] show the relation between the error locator polynomial Λ(x) and the syndromes $S_{i}$. There have been many algorithms like Berlekamp–Massey (BM), Peterson, and others proposed to solve the key equation [3,7], but the inversion-less BM (iBM) algorithm is predominantly used in high throughput architectures [6,21]. Park et al. [22] proposed a novel folded method to reduce the area in the hardware architecture, but the proposed method takes more clock cycles and is proportional to the folding factor. For the final step, the Chien Search (CS) algorithm is used to locate the error position from the error locator polynomial equation. Yoo et al. proposed a low power and high throughput parallel CS algorithm in [23].

### 2.3. Motivation

This paper intends to propose a solution that can address two configurable parameters of a BCH decoder. First, the solution should be scalable across different GF fields, i.e., it should be able to support different GF field extensions ($GF(2^{m})$). Second, the solution should be able to scale across different bit errors *t*. Different configurable BCH decoder solutions have been proposed [20,24], but they lack support for both configurable parameters of the BCH decoders. Inspired by the attempt to solve BCH decoders for multiple GF dimensions in [20], we propose an alternate hybrid approach to have a flexible solution. In [20], a hardware solution was proposed to support multiple BCH codes; however, the circuit area increases in order to support multiple GF dimensions because of the dual-mBCH decoders’ requirement. We have proposed a method to share the hardware logic between the BCH encoder and BCH syndrome generator by modifying the encoding method in [14]. In this paper, we extend our previous work by using a programmable modified syndrome generator algorithm and GPU to have a decoder that works with multiple GF dimensions.

## 3. Hybrid Method

We propose a high throughput system that can correct *t* bit errors over different BCH codes (n,k,t), i.e., error correction over different finite field dimensions, using hardware design and GPU kernel routines. Figure 2 depicts the architectural block diagram for our proposed hybrid method. The flash memory interface is a physical interface to a flash memory device, and the host interface is a standard bus interface, which communicates with the GPU. The GPU could either reside inside the host interface (system on chip) or external to the host system. It is important to note that the GPU system in the system is used for dual purposes, i.e., for the graphical display and error correction. Furthermore, in our proposed method, the GPUs are only deployed when there is an error detected in the page, and the GPUs are not used for pages without error. It is assumed that the host system exercises a memory copy routine to transfer data whenever there is an interaction between the host system and the GPU system. The syndrome generation, proposed in [14], is split into two modules: the Syndrome Residual Unit (SRU) and the syndrome kernel. Then, we propose to implement the modules SRU and the FIFO (shaded area) in hardware and to use GPU kernel routines for the modules’ syndrome calculator, key-equation solver, and an error corrector.

### 3.1. Flowchart

Figure 3 depicts the flow of our proposed hybrid method. Initially, the GPUs create a LUT memory for faster GF multiplication; this method has been proven to be faster on GPUs than threads spawning sub-kernel routines [9]. Next, a page read command is initiated to the flash memory interface. The SRU calculates the *t* residuals of the minimal polynomial, while the data are written into the FIFO. If all the residuals are zero, then we conclude that there are no errors detected in the received vector, and the host shall transfer the data to the application layer. If there are non-zero residuals, then the host calls the Syndrome Calculation Kernel (SK) routine to calculate the syndrome and then calls the Key Equation Kernel (KEK) to form the error locator polynomial Λ(x). Once the Λ(x) is formed, the Chien Search Kernel (CSK) is executed for each bit location. The final error vector is then added to the data in the FIFO to correct the bit errors and then copied to the host memory. Until all the intended data from the flash memory are read, we repeat the previously mentioned steps (Node 1 in Figure 3).

### 3.2. Modified Syndrome Generator

The conventional syndrome generator as discussed in Section 2.2 can be split into two steps: First, the residual polynomial $res_{i}\left(x\right)$, of the received code word r(x), is generated by the equation $res_{i}\left(x\right)=r\left(x\right)\;mod\;\phi_{i}\left(x\right)$, where the minimal polynomial $\phi_{i}\left(x\right)=g_{i,m}x^{m}+\ldots +g_{i,0}$ and residual polynomial $res_{i}\left(x\right)=res_{i,m-1}x^{m-1}+\ldots +res_{i,0}$. Second, the syndrome can be calculated by substituting the primitive element $\alpha^{i}$ in the residual polynomial and is expressed as:(6)$$S_{i}=\sum_{k=0}^{deg(\phi_{i}\left(x\right))-1}res_{i,k}.{\left(\alpha^{i}\right)}^{k}$$

It is clear that by splitting the syndrome generation, $res_{i}\left(x\right)$ does not have any dependency on the field extensions. In fact, the polynomial division used in $res_{i}\left(x\right)$ is identical to a Linear Feedback Shift Register (LFSR) with its coefficient from $\phi_{i}\left(x\right)$. We introduce the idea to have the coefficients of the LFSR as programmable. Figure 4 represents a hardware realization of the SRU array in a serial fashion with programmable feedback coefficients. In most cases, depending on the data interface width of the flash memory interface, we can unfold the serial interpretation of the SRU to process more bits in parallel. Because of the relationship between the conjugacy class and $\phi_{i}\left(x\right)$ [7], it is sufficient to generate *t* SRU units. These SRUs can compute $res_{i}\left(x\right)$ of different minimal polynomials in tandem. Once the residuals are computed, the values of the $res_{i}\left(x\right)$ are compared for non-zero values. An error is triggered if any of the $res_{i}\left(x\right)$ has non-zero coefficients in them, and the GPU kernel routines for the other stages of the BCH decoder are executed.

### 3.3. GPU Kernel Routines

Kernel routines are the fundamental sub routines, representing the SIMDtype of parallelism, executed by the GPU for our proposed decoder. Figure 5 illustrates a systematic execution of the kernel routines where PG2 and PG4 represent pages with errors. PG0, PG1, and PG3 represent pages without error. When there are no errors, the latency incurred is the computation time consumed by the SRU systolic array, as shown in Figure 5. Furthermore, to achieve better throughput, the SRU units can compute $res_{i}\left(x\right)$ for the next page, while the GPU kernel routines are triggered during an error scenario. For GF multiplication in the algorithm, the multiplicand and multiplier are converted to the power basis by referring to the LUT in the global memory. Thus, the multiplication is transformed into a simple XOR operation in the power basis domain. After the multiplication is computed, a reverse transformation is performed by referring to another basis converter LUT in the global memory. The three basic GPU kernel routines used in our approach are explained in detail below.

#### 3.3.1. Syndrome Kernel

In this routine, the $S_{i}$ is calculated by substituting the $\alpha_{i}$ in the equation $res_{i}\left(x\right)$. Since there are no dependencies on the syndromes, *t* parallel SK routines are launched within the GPU. Algorithm 1 represents the pseudocode for the syndrome routine. The *atomicXor* operation is required to synchronize the value updated by the SK routines across multiple threads.

 **Algorithm 1** Syndrome kernel.
1:**procedure**synd kernel($res_{i},S_{i}$)2:  sum←03:  **for**
$j\leftarrow 0,deg(\phi_{i}\left(x\right)-1$
**do**4:    $sum\leftarrow sum+res_{i,j}.\alpha^{i.j}$5:  **end for**6:  *atomicXor($S_{i},sum$)*               ▹ synchronize between threads7:
**end procedure**



#### 3.3.2. Key-Equation Kernel

The KEK is the only single thread routine, in our proposal, because of the iterative nature of the iBM algorithm. Algorithm 2 represents the pseudocode for iBM implementation in the GPU routine. There are other methods like the simplified iBM (siBM) algorithm [7] proposed for the key-equation solver module, but experimental results have proven that siBM does not have significant improvement on the performance of the GPU kernel routines.

 **Algorithm 2** Key-equation kernel.
1:**procedure**keq eq kernel(Λ,S)2:   $\Lambda^{\left(0\right)}\leftarrow 1+S_{1}x$3:   **if**
$S_{1}=0$
**then**4:   $d_{p}\leftarrow 1;\beta^{\left(1\right)}\leftarrow x^{3};l_{1}\leftarrow 0$5:   **else**6:   $d_{p}\leftarrow S_{1};\beta^{\left(1\right)}\leftarrow x^{2};l_{1}\leftarrow 1$7:   **end if**8:   **for**
r←1,t−1
**do**9:   $d_{r}\leftarrow \sum_{i=1}^{t}\Lambda_{i}^{\left(r\right)}S_{2r-i+1}$10:   $\Lambda^{\left(r\right)}\leftarrow d_{p}\Lambda^{\left(r-1\right)}+d_{r}\beta^{\left(r\right)}$11:   **if**
$d_{r}=0\;or\;r<l_{r}$
**then**12:    $\beta^{\left(r+1\right)}\leftarrow x^{2}\beta^{\left(r\right)};l_{r+1}\leftarrow l_{r};d_{p}\leftarrow d_{p}$13:   **else**14:    $\beta^{\left(r+1\right)}\leftarrow x^{2}\Lambda^{\left(r\right)};l_{r+1}\leftarrow l_{r}+1;d_{p}\leftarrow d_{r}$15:   **end if**16:   **end for**17:
**end procedure**



#### 3.3.3. Chien Search Kernel

The CS algorithm is the final step within the decoder. The primitive element $\alpha^{pos^{-1}}$ is checked if it is a root for the error locator polynomial Λ(x) as specified in [9]. This kernel routine is an ideal candidate for the GPU because of the parallelism it offers. Each element of the finite field is evaluated in the equation Λ(x) as shown in Algorithm 3. This evaluation kernel routine is independent for each GF element; hence these routines can be launched in parallel threads. Similar to the SK routine, the memory within the GPU device is shared between threads, so the *atomicXOR* operation is used to avoid writing overlap by different CSK routines. Once the error vector is formed, the error is masked with the data in the memory to yield corrected data.

 **Algorithm 3** Chien search. Kernel
1:**procedure**Chien kernel(Λ,pos,err)2:   sum←1                                  ▹ Always KEQ is minimal3:   **for**
$j\leftarrow 0,deg(\phi_{i}\left(x\right))-1$
**do**4:   $sum\leftarrow sum+\Lambda_{j}{\left(\alpha^{pos^{-1}}\right)}^{j}$5:   **end for**6:   **if**
sum=0
**then**                                    ▹$\alpha^{pos^{-1}}\;\;is\;\;a\;\;root$7:   *atomicXor(err[pos],sum)*                           ▹ Prevent overlap write8:   **end if**9:
**end procedure**



## 4. Experimental Results and Analysis

The proposed hybrid approach was compared against conventional GPU [8,9] and hardware [20] architectures. The hardware implementation of the syndrome generator was synthesized for 28-nm technology, and it achieved an operational frequency of 1 GHz. The setup used for the GPU implementation is given in Table 1. In our experiments, we analyzed the performance and the area consumed for different BCH code sizes. We have used the finite field dimension of m=12,13,14,15 in our comparison, which corresponds to block sizes of 256,512,1024,2048 bytes. Furthermore, we have analyzed the results for different bit errors (t=2,…,40) in our experiments.

### 4.1. Error Analysis

The error correction capability increased with *n*, but the larger the *n*, the higher the probability of random bit error. Based on the raw bit error probability *p*, parity bits, and message code cpar=2 · *t* · m+|m(x)|, the sector with correctable error $\left(P_{secErr}\right)$ might increase and is given as:(7)PsecErr=1−∑i=t+1cparcpari·pi·(1−p)(cpar)−i

Figure 6 plots the bit error vs. sector error for different BCH codes. We can also see that the $P_{secErr}$ decreased as *p* decreased. Furthermore, there was a slight increase in $P_{secErr}$ when compared against different *m*. This was due to the increase in the probability of error within bigger sector sizes.

### 4.2. Syndrome Generation Analysis

Syndrome generation is the critical area where the proposed hybrid method provides an advantage over the GPU methods [8,9]. Figure 7 shows the plot of the syndrome computation time vs. different bit errors (t=2,…,40) across different finite fields (m=12,13,14,15) for different architectures: GPU [9], hardware [21], and hybrid (proposed). The computation of the SRU engine depended on the number of clock cycles required for a page read. Since all the $res_{i}\left(x\right)$ that were necessary for the key-equation solver were calculated in tandem, the latency only depended on the read cycles for flash memory. The hardware architecture for syndrome generation consumed the same clock cycle as the hybrid approach since the approach to syndrome generation was similar. It should be noted that the GPU unit was used as a display unit, so the results of kernel profiling depended on the load of the GPU during the execution of the kernel routine. The execution time for the syndrome on the GPU architecture depended on the number of threads getting executed, and typically, it was from 30–100 μs.

### 4.3. VLSI Analysis

Table 2 compares the hardware area required for different methods of the syndrome generator ([20] and the proposed). Since the GPUs were employed for key-equation and Chien search, the hardware was only compared for the syndrome generator unit for multi-bit error correction for different GF dimensions. This is a fair comparison since the area of the GPUs was already accounted for systems with graphical display. The hardware implementations were targeted for 28-nm and met a frequency of 1 GHz. In order to support different finite field (m=12,13,14,15) and 40-bit error correction, the hardware method [20] consumed 30,247 μm${}^{2}$, whereas our proposed hybrid architecture consumed 10,633 μm${}^{2}$, thus saving two-fold of the area. This area savings was due to the splitting of the syndrome generation into two units (SRU and syndrome kernel). When there were no errors in the page, the total time taken by the proposed decoder was less than 5 μs (Figure 7), which was less than the average read latency of 100 μs. Table 3 compares the power consumed by the conventional method [20] and our proposed method. The last entry in the table provides the power consumption required to support error correction over different fields (m=12,13,14,15) and until 40-bit error correction. There was a savings of 4 mW in our proposed method.

### 4.4. Performance Analysis

The comparison of the total time taken, in case of an error, is compared for the hardware [20], GPU [8,9], and hybrid architecture (proposed) in Figure 8 (the variable *be* represents bit error). We can see that the hybrid approach was better than the GPU method because of the SRU implementation in hardware. However, the hardware implementation took less than 1 μs because of the high performance (which is indistinguishable in Figure 8). We can observe a gain of more than 25% when the system has errors. We can find the probability of sector error from a given bit error rate using Equation (Equation 7). For a given $P_{secError}$, the throughput was calculated for a second’s worth of data transfer.

Figure 9 represents the plot for throughput vs. bit error rate for different finite fields (m=12,13,14,15) and different t=4,…,40. We can see that for m=12,13, the throughput was sustained till $10^{-3}$, and for m=14,15, the throughput was sustained till $10^{-3.5}$. This throughput is sustainable for flash memories that have an Uncorrectable Bit Error Rate (UBER) of $10^{-15}$, an it is also sustainable for the end of life for flash memories which is greater than $10^{-5}$.

## 5. Conclusions

In this paper, we have proposed a novel hybrid method to implement an efficient BCH decoder for different finite field extensions by having the SRU module in hardware and the rest implemented in GPU kernel routines. By using this method, we have given flexibility on two parameters: first, the flexibility over different finite fields $GF(2^{m})$; second, the flexibility over different bit error support. The flexibility over $GF(2^{m})$ was achieved by splitting the syndrome into the SRU unit and the syndrome kernel. The SRU module resided on the Euclidean domain of GF(2) polynomials, thus making it programmable across multiple finite fields. The syndrome kernel was executed only when an error was encountered. The latency taken by our method, without error, was superior to the CPU and GPU implementations and was equal to the performance observed in [20]. Besides, we had two-fold area savings in the SRU unit to achieve flexibility over $GF(2^{m})$ and bit errors. Therefore, this hybrid approach is a pragmatic solution to have a flexible error correction for modern NAND flash devices.

## Figures and Tables

**Figure 1 micromachines-10-00365-f001:**
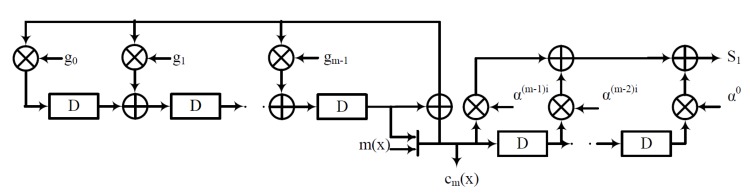
Area-efficient syndrome generator.

**Figure 2 micromachines-10-00365-f002:**
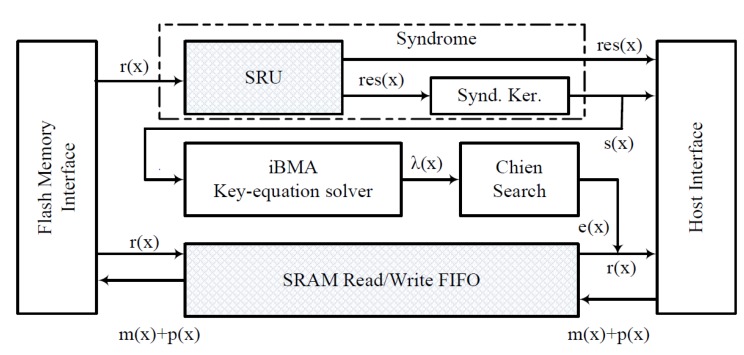
Hybrid BCH decoder block diagram.

**Figure 3 micromachines-10-00365-f003:**
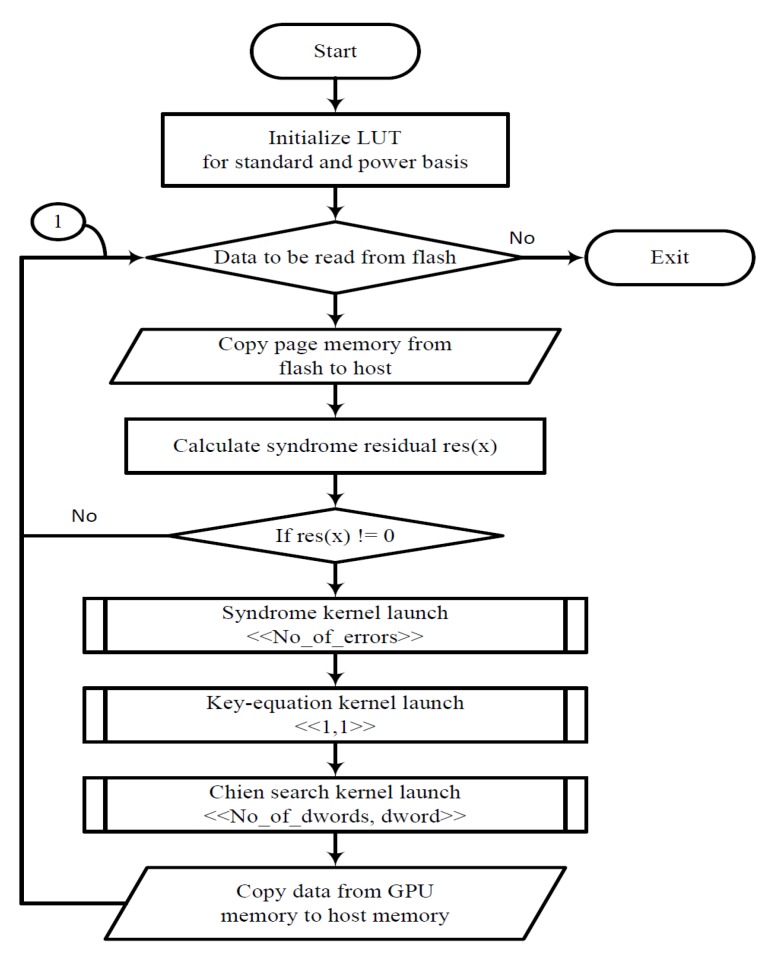
Flow chart for the hybrid system.

**Figure 4 micromachines-10-00365-f004:**
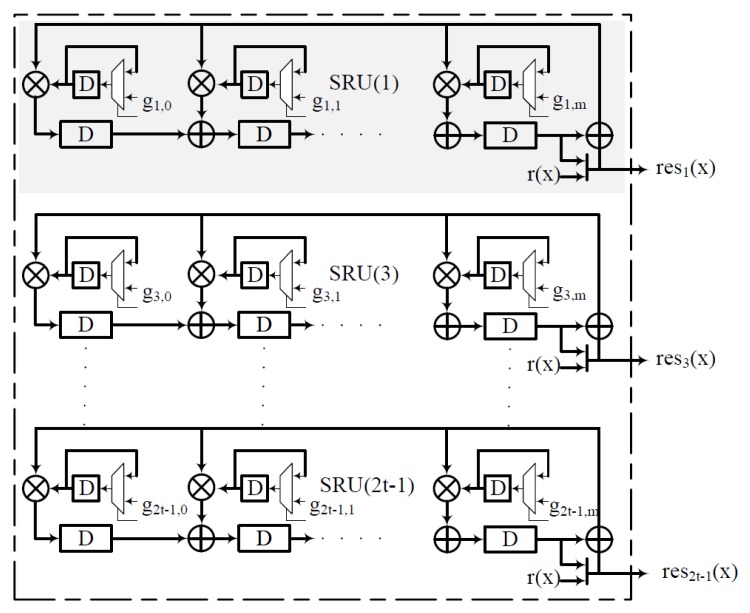
Array of the syndrome residual unit.

**Figure 5 micromachines-10-00365-f005:**
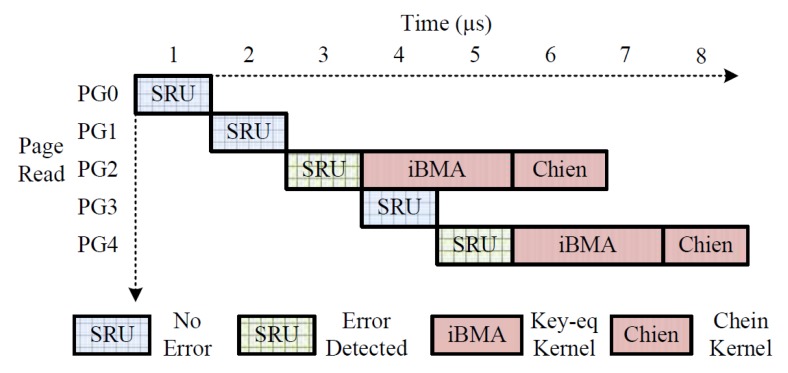
Decoder execution sequence. PG, Page.

**Figure 6 micromachines-10-00365-f006:**
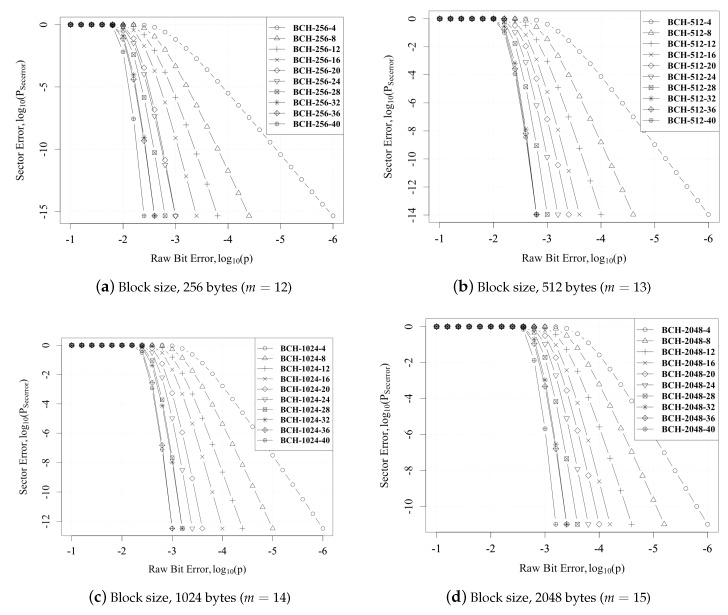
Raw bit error vs. sector error.

**Figure 7 micromachines-10-00365-f007:**
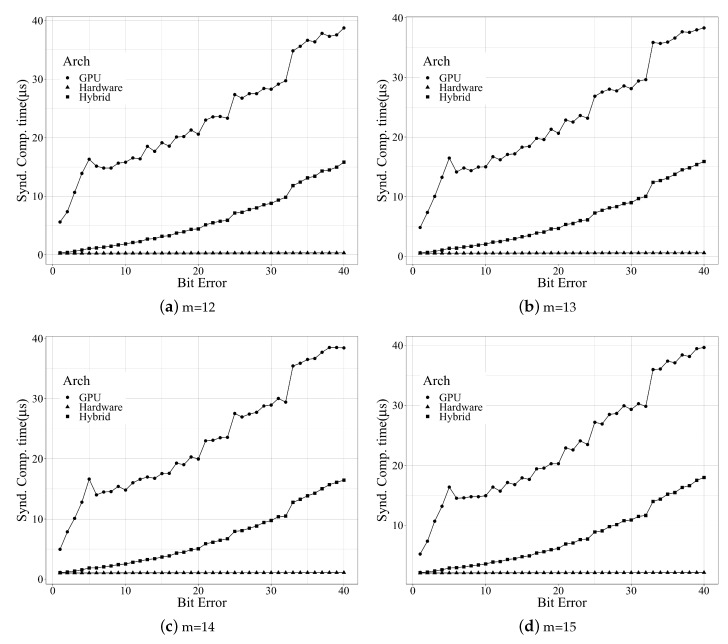
Syndrome computation time for different arches.

**Figure 8 micromachines-10-00365-f008:**
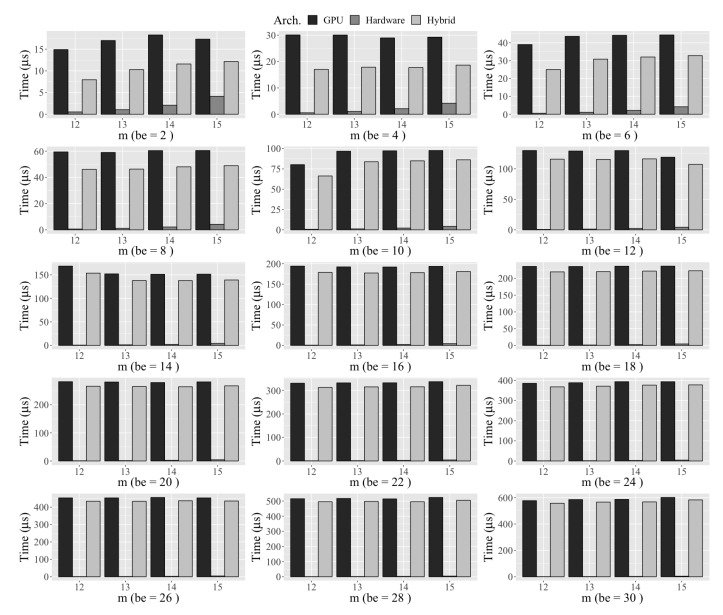
Total computation time for different architectures and different finite fields.

**Figure 9 micromachines-10-00365-f009:**
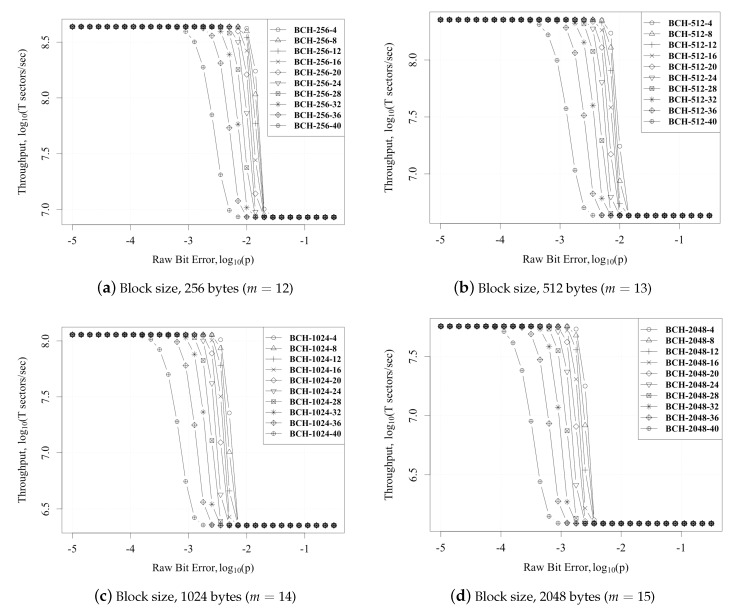
Raw bit error vs. throughput (sectors/s).

**Table 1 micromachines-10-00365-t001:** Experimental setup.

	GPGPU	CPU
Platform	Geforce GTX 760. 1152 cores	Intel Xeon i7
Clock Freq.	1.033 GHz	3.7 GHz
Memory	GDDR5(2 GB), 6 Gbps	DDR2 (32 GB), 102.4 Gbps

**Table 2 micromachines-10-00365-t002:** Area comparison for different syndrome generators vs. the proposed SRU.

	Setup	Area for *t* (μm${}^{2}$)									
		*4*	*8*	*12*	*16*	*20*	*24*	*28*	*32*	*36*	*40*
**m =** ***12***	[20]	606	1287	2079	2871	3256	3661	3959	4252	4611	4917
	Prop.	853	1704	2550	3399	4209	5061	5903	6747	7592	8436
**m =** ***13***	[20]	858	1863	2962	4043	4648	5246	5814	6387	6988	7573
	Prop.	924	1846	2767	3682	4607	5532	6390	7305	8308	9134
**m =** ***14***	[20]	916	2002	3167	4330	5016	5723	6375	7048	7714	8390
	Prop.	994	1985	2976	3966	4966	5959	6948	7942	8935	9927
**m =** ***15***	[20]	988	2172	3435	4707	5485	6292	7043	7818	8587	9367
	Prop.	1064	2125	3186	4249	5318	6383	7448	8504	9570	10,633
**m =** ***12, …, 15***	[20]	3368	7324	116,43	15,951	18,405	20,922	23,191	25,505	27,900	***30,247***
	Prop.	1064	2125	3186	4249	5318	6383	7448	8504	9570	***10,633***

**Table 3 micromachines-10-00365-t003:** Power comparison for different syndrome generators vs. the proposed SRU.

	Setup	Power for *t* (mW)									
		*4*	*8*	*12*	*16*	*20*	*24*	*28*	*32*	*36*	*40*
**m =** ***12***	[20]	0.179	0.374	0.599	0.819	0.897	0.978	1.037	1.097	1.170	1.232
	Proposed	0.167	0.336	0.499	0.673	0.841	1.014	1.185	1.354	1.524	1.695
**m =** ***13***	[20]	0.226	0.491	0.773	1.054	1.178	1.304	1.426	1.546	1.674	1.797
	Proposed	0.178	0.355	0.534	0.714	0.889	1.061	1.248	1.424	1.615	1.778
**m =** ***14***	[20]	0.231	0.503	0.784	1.068	1.213	1.363	1.499	1.640	1.780	1.923
	Proposed	0.187	0.373	0.561	0.747	0.938	1.122	1.309	1.493	1.678	1.863
**m =** ***15***	[20]	0.235	0.512	0.812	1.110	1.270	1.438	1.593	1.755	1.915	2.078
	Proposed	0.194	0.388	0.588	0.776	0.975	1.168	1.370	1.564	1.755	1.953
**m =** ***12, …, 15***	[20]	0.692	1.506	2.369	3.232	3.661	4.105	4.518	4.941	5.369	***5.798***
	Proposed	0.194	0.388	0.588	0.776	0.975	1.168	1.370	1.564	1.755	***1.953***

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
