# Peer review of "A Flexible Hybrid BCH Decoder for Modern NAND Flash Memories Using General Purpose Graphical Processing Units (GPGPUs)"

_micromachines, 2019, doi:10.3390/mi10060365_

Round 1

Reviewer 1 Report

The manuscript (micromachines-499531) shows interesting results of A flexible hybrid BCH decoder for modern NAND Flash memories using GPGPUs. It reports a comprehensive analysis and results in detail electrically. Some minor comments would like to provide here from this referee and hope to improve the content for general readers: 

Authors should provide the abbreviation in detail, such as BCH and LDPC etc. Although referee and expect in the same area would like the terminology, it would be good for general readers if authors can provide in detail for them. Please review the whole manuscript and provide in detail. 

It is quite good authors provide the time- and design area- related parameters and results for their proposal architecture as compared to others. How about the energy or power consumption in the benchmark of comparison?

It would be good if authors can provide a sort of benchmark (time/design area/power consumption) as compared to recently proposed architectures (as authors mentioned in introduction part with different group studies). That would be quite useful to understand the key contribution in this work as compared to recently NAND flash error correction technique progress. Table or Figure would be helpful since they may use different technology node or operation frequency as comparison. 

Author Response

Point 1: The manuscript (micromachines-499531) shows interesting results of A flexible hybrid BCH decoder for modern NAND Flash memories using GPGPUs. It reports a comprehensive analysis and results in detail electrically. Some minor comments would like to provide here from this referee and hope to improve the content for general readers:

Authors should provide the abbreviation in detail, such as BCH and LDPC etc. Although referee and expect in the same area would like the terminology, it would be good for general readers if authors can provide in detail for them. Please review the whole manuscript and provide in detail. 

Response:  We have addressed the abbreviation detail and have included the frequently used abbreviation in the section “Abbreviations”. We thank the reviewer for this suggestion.

Point 2: It is quite good authors provide the time- and design area- related parameters and results for their proposal architecture as compared to others. How about the energy or power consumption in the benchmark of comparison?

Response:  We have presented results of power analysis, as suggested, in Table 3 (newly added) against [20] for different bit errors and different Galois Field (GF).

Point 3: It would be good if authors can provide a sort of benchmark (time/design area/power consumption) as compared to recently proposed architectures (as authors mentioned in introduction part with different group studies). That would be quite useful to understand the key contribution in this work as compared to recently NAND flash error correction technique progress. Table or Figure would be helpful since they may use different technology node or operation frequency as comparison. 

Response: Table 2 and 3 compares the area and power against reference [20] which uses flexibility for different GF. The key contribution in this paper is the flexibility provided for different GF and different bit error rates with high throughput. We used 28-nm as the target technology for the reference [20] and proposed.

Response: We hope the revised paper addresses the concerns. Here is the summary of the changes in this revised version

1.     Section 2 (Background): We have added more references to the background section and describe more on the previous work related to our paper.

2.     Section 3, Figure 2 has minor changes. The highlighted syndrome area is computed in hardware and kernel routine. This is one of our key contributions, which uses equation 6.

3.     Section 4: This section has major changes with new results

a.     Figure 6 shows the BER vs. Section error analysis

b.    Figure 7 has results for 40-bit error, and the results for different block sizes are separated to have better visibility on our hybrid approach

c.     Table 2 & 3 are revised for m=12,..15 and t=4,..,40

d.    Figure 8 and 9 show the performance analysis for different m and t

Reviewer 2 Report

The paper presented by the authors deals with the acceleration of adaptive BCH codes for NAND Flash memories on GPGPUs. Although the work is well presented in some sections, I am really having serious doubts on the application of this strategy in state-of-the-art NAND Flash technology. The first reason is that the Bit Error Rate of nowadays 3D NAND Flash memories is so high that makes BCH codes useless (all ECC are based on LDPC now). Second of all, the correction latencies indicated in the simulations are not in line with the requirements of typical storage applications like Solid State Drives or mobile computing.

In the present form I cannot recommend this work for publication.

Author Response

Point 1: The paper presented by the authors deals with the acceleration of adaptive BCH codes for NAND Flash memories on GPGPUs. Although the work is well presented in some sections, I am really having serious doubts on the application of this strategy in state-of-the-art NAND Flash technology.

Response: The applications that we propose targets systems with GPUs already in it. For example, the GPUs in Personal Computers can be used for error correction. Also, mobile devices have GPUs embedded with their process; for example, the A12X processor has GPUs that could be programmed using OpenACC tool https://en.wikipedia.org/wiki/Apple_A12X. Our motivation is to use these GPUs already available for error correction.

Point 2:  The first reason is that the Bit Error Rate of nowadays 3D NAND Flash memories is so high that makes BCH codes useless (all ECC are based on LDPC now).

Response: The BER is around 10^{-4} for modern NAND flash devices, and our method could sustain such BER in throughput.Figure 7 shows the BER vs throughput for our proposed solution. Modern memories come with two options 1. To have ECC within memory device 2. To have ECC outside memory device. Most ECC within the flash device choose to have LDPC, but the focus of our research is on option 2.

Point3: Second of all, the correction latencies indicated in the simulations are not in line with the requirements of typical storage applications like Solid State Drives or mobile computing.

Response: Since Figure 6 in the previous version had all the computation in a single graph, it was difficult to distinguish between the hardware and hybrid approach. We have revised the simulation results for 40 bit error correction and have split the syndrome computation time for different block sizes (Figure 7 in the revised version). In this, the hybrid approach latency is less than 20 us for 40-bit errors, and this is less than the NAND flash typical read latency of 20us (for page Read).

Point4: In the present form I cannot recommend this work for publication.

Response: We hope the revised paper addresses the concerns. Here is the summary of the changes in this revised version

1.     Section 2 (Background): We have added more references to the background section and describe more on the previous work related to our paper.

2.     Section 3, Figure 2 has minor changes. The highlighted syndrome area is computed in hardware and kernel routine. This is one of our key contributions, which uses equation 6.

3.     Section 4: This section has major changes with new results

a.     Figure 6 shows the BER vs. Section error analysis

b.    Figure 7 has results for 40-bit error, and the results for different block sizes are separated to have better visibility on our hybrid approach

c.     Table 2 & 3 are revised for m=12,..15 and t=4,..,40

d.    Figure 8 and 9 show the performance analysis for different m and t

Reviewer 3 Report

The paper presents a decoder design for BCH codes with applications to NAND flash memories. This is a well-studied topic and the references in the paper are not comprehensive. The proposed approach is based on a hybrid decoding strategy where the syndrome calculation is implemented in hardware, whereas the BMA and Chien search are executed in software on a Graphical Processing Unit (GPU). I am not familiar with GPU implementations. There may be some novelty in the presented design. However, the concept of hybrid decoding is not new. Probably, most of the early ECC units based on Reed-Solomon codes employed such a hybrid approach, where the syndrome is calculated on the fly in hardware while the data is read from the flash. In case of a detected error, the actual error correction is performed in software on a CPU. This approach is suitable for low channel error rates, where there is a high probability of error free codewords. Todays flash memories require long (GF(2^14) or GF(2^15)) and strong BCH codes (t=20,…120), where the probability of error free code words is very low at the end of life (EOL). The proposed design would limit the data throughput. The applicability of the proposed approach is doubtful. I do not think that the paper is suitable for this journal.

The authors should improve the references. For instance, here are some additional references for hardware implementations:

X. Zhang and K. K. Parhi, “High-speed architectures for parallel long BHC encoders,”

in IEEE Transactions on very large scale integration (VLSI) Systems, Vol. 13, No. 7,

Jul. 2005.

Sun, K. Rose and T. Zhang, “On the Use of Strong BCH Codes for Improving Multilevel NAND Flash Memory Storage Capacity,” IEEE Workshop on Signal Processing Systems (SiPS), Oct. 2006

W. Liu, J. Rho, and W. Sung, “Low-power high-throughput BCH error correction

VLSI design for multi-level cell NAND flash memories,” in IEEE Workshop on Signal

Processing Systems Design and Implementation (SIPS), oct. 2006, pp. 303 –308.

F. Sun, S. Devarajan, K. Rose, and T. Zhang, “Design of on-chip error correction

systems for multilevel NOR and NAND flash memories,” Circuits, Devices Systems,

IET, vol. 1, no. 3, pp. 241 –249, June 2007.

C. Wang, Y. Gao, L. Han, and J. Wang, “The design of parallelized BCH codec,” 3rd

International Congress on Image and Signal Processing (CISP), 2010.

J. Freudenberger, J. Spinner, “A configurable Bose-Chaudhuri-Hocquenghem codec architecture for

flash controller applications”, Journal of Circuits, Systems and Computers, Vol. 23, No. 02, 1450019 (2014)

Furthermore, the authors should present throughput results versus bit error probability, because the average throughput with the proposed method will strongly depend on the channel error rate.

Author Response

Point 1: The paper presents a decoder design for BCH codes with applications to NAND flash memories. This is a well-studied topic and the references in the paper are not comprehensive.

Response: Section 2 (Background): We have added more references to the background section and provided more descriptions of the previous work related to our paper.

Point 2: The proposed approach is based on a hybrid decoding strategy where the syndrome calculation is implemented in hardware, whereas the BMA and Chien search are executed in software on a Graphical Processing Unit (GPU). I am not familiar with GPU implementations. There may be some novelty in the presented design. However, the concept of hybrid decoding is not new. Probably, most of the early ECC units based on Reed-Solomon codes employed such a hybrid approach, where the syndrome is calculated on the fly in hardware while the data is read from the flash.

Response: The hybrid approach previously described have the syndrome generator in hardware, but we split the syndrome into two sub-modules SRU and syndrome kernel. We have updated Figure 2 to highlight the syndrome generation. The technique used for the syndrome generation is using the residual method as described in Eq. 6. The other references, for example,

J. Freudenberger, J. Spinner, “A configurable Bose-Chaudhuri-Hocquenghem codec architecture for flash controller applications”, Journal of Circuits, Systems and Computers, Vol. 23, No. 02, 1450019 (2014)

has syndrome calculation using Eq. 5 on page 6. This method has a dependency on the primitive element of the Galois field, so to support multiple GF the syndrome generator has to be replicated. Our proposed hybrid method using modulo arithmetic hence removing the dependency on the GF, since the modulo arithmetic is performed in the Euclidean domain of GF(2) and not it’s extension field.

Point 3: In case of a detected error, the actual error correction is performed in software on a CPU. This approach is suitable for low channel error rates, where there is a high probability of error free codewords.

Response: When an actual error occurs the error correction is performed on the GPUs. The GPUs can run thousands of threads in parallel and this is one of the reasons why we see better performance than CPUs (refer to [8] and [9] in the revised document). We have computed the throughput vs BER for hybrid implementation and we see that it can sustain throughput for BER < 10 ^ {-3}

 Point 4: Todays flash memories require long (GF(2^14) or GF(2^15)) and strong BCH codes (t=20,…120), where the probability of error-free code words is very low at the end of life (EOL). The proposed design would limit the data throughput.

Response: We have updated our results for area/power/BER/throughput for block sizes 256, 512, 1024, 2048 (m=12, 13, 14, 15) and computed till bit error support of 40. The typical BER for flash memories are 10^{-15}, but near EOL this could be 10^{-5}. Our throughput results show that the hybrid approach could sustain until 10^{-4}. This is due to the parallel approach aided by the GPUs on the Chien search method.

Point 5: The applicability of the proposed approach is doubtful. I do not think that the paper is suitable for this journal

Response: The applications that we propose targets systems with GPUs already in it. For example, the GPUs in Personal Computer can be used for error correction. Also, mobile devices have GPUs embedded with their process; for example, the A12X processor has GPUs that could be programmed using OpenACC tool https://en.wikipedia.org/wiki/Apple_A12X. Our motivation is to use these GPUs already available for error correction. We think the modern ECC for NAND flash memories could incorporate GPUs to correct them since they are available in most electronic devices.

Response: We hope the revised paper addresses the concerns. Here is the summary of the changes in this revised version

1.     Section 2 (Background): We have added more references to the background section and describe more on the previous work related to our paper.

2.     Section 3, Figure 2 has minor changes. The highlighted syndrome area is computed in hardware and kernel routine. This is one of our key contributions, which uses equation 6.

3.     Section 4: This section has major changes with new results

a.     Figure 6 shows the BER vs. Section error analysis

b.    Figure 7 has results for 40-bit error, and the results for different block sizes are separated to have better visibility on our hybrid approach

c.     Table 2 & 3 are revised for m=12,..15 and t=4,..,40

d.    Figure 8 and 9 show the performance analysis for different m and t

Round 2

Reviewer 2 Report

no comments

Reviewer 3 Report

I carefully read the authors' responses in which my concerns are clearly resolved. Therefore, I think it can be considered as a publication.